# A pH-sensitive motif in an outer membrane protein activates bacterial membrane vesicle production

Ruchika Dehinwal [1,2], Tata Gopinath[3], Richard D. Smith[4], Robert K. Ernst [4], Dieter M. Schifferli [5] ✉, Matthew K. Waldor [1,2] ✉ & Francesca M. Marassi [3] ✉

Outer membrane vesicles (OMVs) produced by Gram-negative bacteria have key roles in cell envelope homeostasis, secretion, interbacterial communication, and pathogenesis. The facultative intracellular pathogen *Salmonella* Typhimurium increases OMV production inside the acidic vacuoles of host cells by changing expression of its outer membrane proteins and modifying the composition of lipid A. However, the molecular mechanisms that translate pH changes into OMV production are not completely understood. Here, we show that the outer membrane protein PagC promotes OMV production through pH-dependent interactions between its extracellular loops and surrounding lipopolysaccharide (LPS). Structural comparisons and mutational studies indicate that a pH-responsive amino acid motif in PagC extracellular loops, containing PagC-specific histidine residues, is crucial for OMV formation. Molecular dynamics simulations suggest that protonation of histidine residues leads to changes in the structure and flexibility of PagC extracellular loops and their interactions with the surrounding LPS, altering membrane curvature. Consistent with that hypothesis, mimicking acidic pH by mutating those histidine residues to lysine increases OMV production. Thus, our findings reveal a mechanism for sensing and responding to environmental pH and for control of membrane dynamics by outer membrane proteins.

Gram-negative bacteria release OMVs into the environment as a mechanism for secretion and maintenance of cell envelope homeostasis[1,2]. OMVs are spherical nanometer-sized structures that originate from the bacterial outer membrane (OM), and thus have a lipid bilayer OM composed of lipopolysaccharide (LPS) phospholipids and proteins, and a lumen enriched in periplasmic components. OMVs are implicated in diverse processes fundamental for microbe-host interactions, including quorum sensing, interbacterial killing, biofilm formation, virulence, toxin delivery, immune evasion, and colonization[3–5]. Moreover, OMVs provide an attractive platform for biotechnology applications, including vaccine development and drug delivery[2,6].

Despite their importance, a comprehensive mechanistic understanding of OMV biogenesis remains elusive. OMV formation is generally thought to be induced by changes in the composition and physical properties of the OM and cell envelope. Models of OMV biogenesis posit that OM vesiculation can be driven by various factors such as localized accumulation of peptidoglycan fragments and misfolded proteins in the periplasmic space that exert outward turgor pressure[7], accumulation of phospholipids in outer leaflet of the OM[8],

[1]Division of Infectious Diseases, Brigham and Women's Hospital, Boston, USA. [2]Department of Microbiology, Harvard Medical School, Howard Hughes Medical Institute, Boston, MA, USA. [3]Department of Biophysics, Medical College of Wisconsin, Milwaukee, WI, USA. [4]Department of Microbial Pathogenesis, School of Dentistry, University of Maryland, Baltimore, MD, USA. [5]Department of Pathobiology, School of Veterinary Medicine, University of Pennsylvania, Philadelphia, PA, USA. ✉e-mail: dmschiff@vet.upenn.edu; mwaldor@bwh.harvard.edu; fmarassi@mcw.edu

weakening of the links between the OM and peptidoglycan layer[9], and LPS modifications that affect OM physical properties[10–14]. These models are not mutually exclusive and may be active in specific microbes or environmental conditions.

Here we studied OMV biogenesis in *Salmonella enterica* subsp. *enterica* serovar Typhimurium (STm), a facultative intracellular pathogen that causes enterocolitis in humans and typhoid-like disease in susceptible mice[15]. To enhance its intracellular survival STm activates the two-component system PhoP-PhoQ (PhoPQ), an environmental sensor triggered by changes in $Ca^{2+}$ and $Mg^{2+}$ concentration, pH, and cationic antimicrobial peptides[16,17]. PhoPQ stimulation upregulates *pag* (PhoP-activated genes) critical for survival within macrophages and pathogenesis[18–20]. Pag proteins promote both LPS modifications and OMV biogenesis, with the latter providing a mechanism for shedding unmodified LPS to accommodate newly modified LPS in the bacterial OM[21]. These activities are largely attributed to three outer membrane proteins: PagL, PagP, and PagC. PagL and PagP are enzymes that modify the lipid A moiety of LPS[22,23], a process that has been proposed to promote OMV formation[11–13]. PagC is a 162-residue protein that increases OMV formation[24,25] but the mechanism by which PagC activates OM vesiculation is unknown.

Here, we show that PagC alone is sufficient for enhancing STm OMV production, and we identify a pH-responsive amino acid motif in its extracellular loops responsible for vesiculation. Using lipidomics, structure-guided mutational analyses, cell-based assays of OMV production, and molecular dynamics (MD) simulations, we show that PagC does not chemically modify LPS, but instead responds to mild acidification by altering the three-dimensional structure and flexibility of its extracellular loops and their interactions with the surrounding LPS. These changes can alter membrane curvature leading to OM vesiculation. Our findings provide a perspective on how outer membrane proteins can modulate OMV formation.

## Results and discussion

### PagC expression does not modify the lipid A moiety of LPS

PagL removes and PagP adds an acyl chain from the lipid A moiety of LPS (Fig. 1A)[22,23]. The resulting change in the ratio of the size of the polar headgroup to acyl chains is thought to induce a conical LPS structure that promotes OM curvature and facilitates OMV budding[11–13]. PagC, on the other hand, has been shown to promote OM vesiculation independently of PhoPQ-regulated LPS remodeling, since its expression in a *phoPQ* deletion mutant is sufficient to increase OMV production[24,25].

To examine whether PagC, chemically modifies STm LPS, we compared lipid A isolated from either wild-type STm or a *pagC* deletion mutant (Δ*pagC*) grown under PhoPQ-inducing conditions (5.8L media: pH 5.8, 10 μM $Mg^{2+}$) or non-PhoPQ-inducing conditions (7.6H media: pH 7.6, 10 mM $Mg^{2+}$), using mass spectrometry and gas chromatography. LPS isolated from either wild-type or Δ*pagC* STm were similar (Fig. 1B), indicating that PagC does not affect the acylation profile of lipid A. Similarly, no differences in lipid A acyl chain modifications were detected in LPS purified from OMVs produced by wild-type or Δ*pagC* bacteria grown in 5.8L, 7.6H, or LB media (Fig. 1C). In addition, the LPS O-antigen isolated from wild-type and Δ*pagC* strain showed very similar migration patterns on SDS-PAGE (Fig. 1D). Taken together, these results demonstrate that unlike PagP and PagL, PagC does not chemically modify LPS.

### The extracellular loops of PagC are responsible for OMV production

PagC belongs to the Ail/Lom protein family (pfam 06316) and shares high sequence similarity (Fig. S1A) with the other family members, Rck from STm (53% identity) and Ail from *Yersinia pestis* (36% identity). Notably, however, neither Ail nor Rck are reported to increase OMV production[25]. Indeed, we observed that OMV production, which was highly impaired in Δ*pagC*, was restored by expressing plasmid-

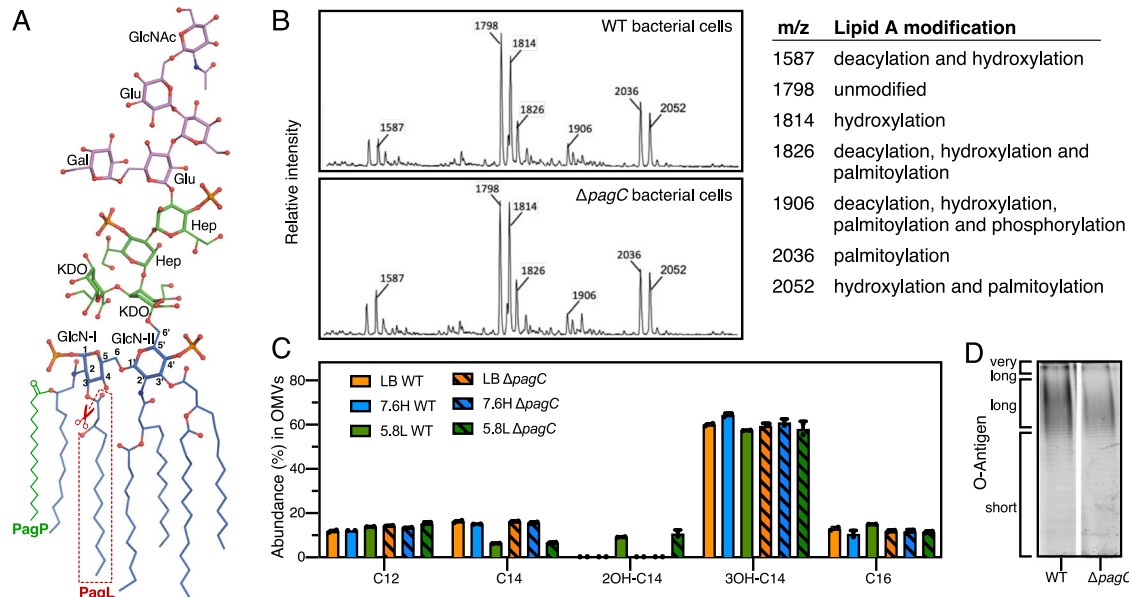

**Fig. 1 | Lipid A modifications of LPS from wild-type and Δ*pagC* STm.** LPS was purified from wild-type and Δ*pagC* bacteria or OMVs, grown in PhoPQ-inducing conditions (5.8L media) non-PhoPQ-inducing conditions (7.6H media) or lysogeny broth (LB media). **A** Structure of the major form of STm LPS showing lipid A (blue) with GlcN (d-glucosamine); inner core (green) with KDO (2-keto-3-deoxy-octulosonic acid) and Hep (l-glycero-d-manno-heptose); outer core (pink) with Glu (d-glucose), Gal (d-galactose); and O-antigen (pink) with GlcNAc (N-acetyl-glucosamine). PagP adds a palmitoyl chain (C16:0) to C2 of GlcN-I (green). PagL deacetylates the C3 of GlcN-I 3 (red). **B** Mass spectrometry analysis of LPS isolated from bacteria grown under

PhoPQ-inducing conditions (*n* = 3 biological replicates). Assignment of *m/z* peaks was based on previous work[58]. **C** Gas chromatography analysis of LPS isolated from OMVs of wild-type and Δ*pagC* STm grown in LB, 7.6H or 5.8L media. Acyl chain moieties correspond to C12, C14, C16, 2OHC14, and 3OHC14. Mean values with standard error of the mean (SEM) are shown (*n* = 2 biological replicates). No statistically significant differences ($p > 0.05$) were found between the groups grown in different conditions, using one-way ANOVA multiple-comparison test. **D** SDS-PAGE analysis of LPS purified from wild-type and Δ*pagC* bacteria, visualized with Pro-Q Emerald LPS gel stain (*n* ≥ 3 biological replicates). Source data are provided as a Source Data file.

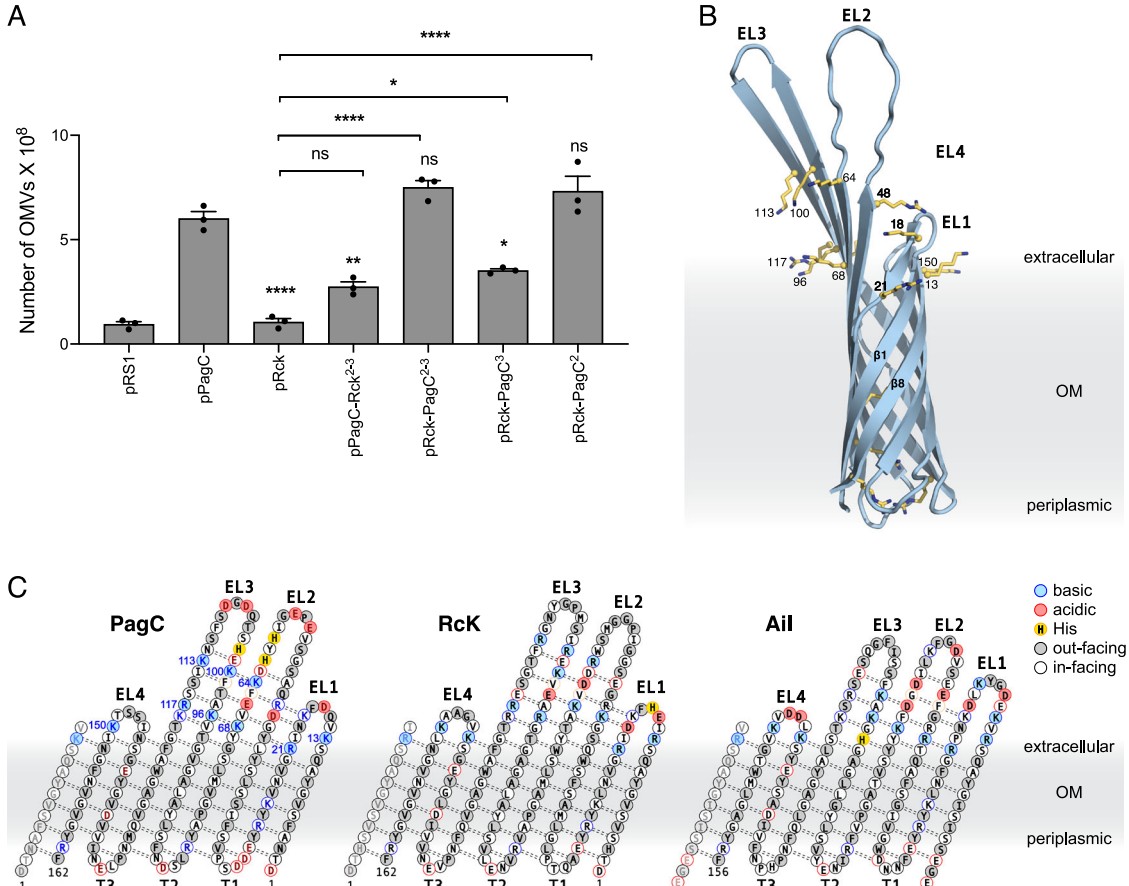

**Fig. 2 | OMV production by PagC-Rck chimeras. A** Number of OMVs produced by pPagC, pRcK, pPagC-Rck chimeras, or pRS1 (empty plasmid) in a Δ*pagC*, Δ*rck*, Δ*ompX*, Δ*pgtE* STm mutant strain. Mean values with SEM are shown (*n* = 3 biological replicates). Statistical significance was calculated using one-way ANOVA multiple-comparison test and set at a *p* < 0.05 (*: *p* < 0.05; **: *p* < 0.01; ns: *p* > 0.05). The *p*-values indicated above the bars represent comparison to pPagC and *p*-values above the brackets represent comparison to pRck. **B** AlphaFold structural model of PagC showing basic side chains (yellow sticks). **C** Sequences of PagC, RcK, and Ail viewed in the context of the canonical eight-stranded β-barrel topology with four extracellular loops (EL1-EL4) and three intracellular turns (T1-T3). Colors reflect sidechains facing the barrel exterior (color-filled) or the barrel interior (white), basic (blue) or acidic (red) sidechains, and His (yellow). Dashed lines represent hydrogen bonds. Source data are provided as a Source Data file.

encoded PagC (pPagC) but not by expressing plasmid-encoded RcK (pRcK), which resulted in similarly low numbers of OMVs as Δ*pagC* harboring the empty plasmid vector (pRS1; Fig. 2A). To understand the differential capacity of these proteins to promote vesiculation, we examined their sequences in the context of the Ail/Lom canonical structure, represented by Ail[26–28] which forms an eight-stranded transmembrane β-barrel with three short intracellular periplasmic turns (T1-3) and four extracellular loops (EL1-4) (Fig. S1B). Structure-based sequence alignment highlighted the pronounced sequence conservation of the three proteins in the transmembrane β-barrel, and sequence divergence in EL1-4 (Fig. 2B, C).

To test whether extracellular loop sequences specific to PagC account for the OMV production activity, we engineered a series of plasmid-encoded PagC-Rck chimeric proteins (Table S1; Fig. S2) where the EL sequences from one protein were replaced with their counterparts from the other. We used a STm mutant strain (Δ*pagC*, Δ*rck*, Δ*ompX*, Δ*pgtE*) lacking *pagC* and its structural and functional homologs, tested it for growth fitness under both PhoPQ -inducing and -non-inducing conditions (Fig. S3) and used it for the expression of the chimeras to analyze their effect on OMV production (Fig. 2A). The strain expressing the PagC-Rck[2-3] chimera (PagC EL2-EL3 replaced by RcK EL2-EL3, pPagC-Rck[2-3]) produced significantly fewer OMVs compared to the isogenic strain expressing PagC (pPagC). In contrast, the strain expressing Rck-PagC[2-3] (Rck EL2-EL3 replaced with PagC EL2-EL3,

pRck-PagC[2-3]) produced at least as many OMVs as observed with pPagC and significantly greater than those detected with pRck. Additional chimeras where EL2 or EL3 of Rck were replaced individually by EL2 or EL3 of PagC, showed that both pRck-PagC[2] and pRck-PagC[3] increase OMV production compared to pRcK, with pRck-PagC[2] yielding OMV levels similar to pPagC (Fig. 2A). Together these results indicate that the EL2 and EL3 sequences of PagC have a potent influence on OMV production, with EL2 having the greatest effect.

Structure-based sequence alignment also revealed the presence of outward-facing basic residues predicted to protrude from the PagC transmembrane barrel near the base of the loops and from the mid-sections of EL2 and EL3 (Fig. 2B, C). Similar basic residues are found at parallel positions in RcK and Ail. In Ail, these residues were shown to be critical for establishing polar contacts with the neighboring LPS, affecting the physical state of the OM, and promoting *Y. pestis* survival and cell envelope integrity[29]. To examine their roles in PagC-activated OMV production, we engineered a set of *pagC* allelic variants, where they were substituted with Ala (Table S1). A PagC mutant with extensive substitutions in EL2 (R48A, H60A, H62A, K64A, K68A) or EL3 (K96A, K100A, H102A, K113A, R117A, K118A) failed to export to the OM, and produced similar, low levels of OMVs as the Δ*pagC* strain, suggesting that these substitutions abolish recognition by the OM biogenesis apparatus and proper folding[30]. The less extensively substituted PagC mutants, however, were detectable in the OM and

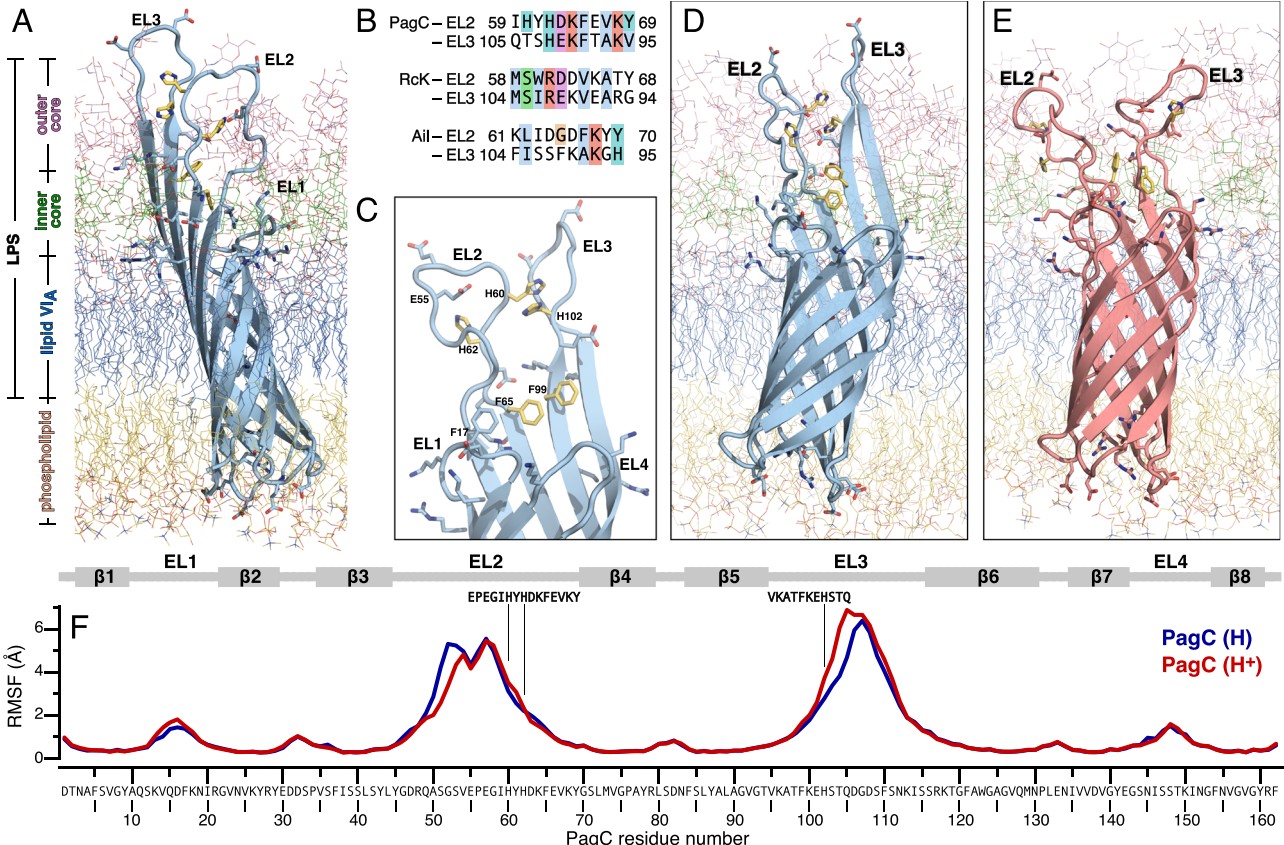

**Fig. 3 | Conformation of the pH response motif of PagC. A** Representative structure (taken at 3 μs) of one independent MD simulation of His-neutral PagC in the STm OM. **B** Sequence alignment of the ascending and descending neighboring segments of EL2 and EL3 rendered with ClustalX coloring using Jalview[59]. MD simulations of His-neutral (**C, D**; 2394 ns) or His-protonated (**E**; 2668 ns) PagC. Models show key His and Phe sidechains (yellow stick) and basic and acidic sidechains (stick). The OM outer leaflet is LPS (blue, green, pink lines) and the inner leaflet is phospholipid (yellow lines). **F** Time-averaged RMSF of PagC residues calculated for Carbon atoms. Each trace is the average over the last 2 μs of three independent 3 μs MD simulations of His-neutral (blue) or His-protonated (red) PagC. Protein secondary structure is shown above the data. Source data are provided as a Source Data file.

produced similarly low numbers of OMVs as ΔpagC bacteria (Fig. S4A). Overall, the data suggest that outward-facing basic residues in EL2 and EL3 of PagC play an important role in OMV production.

### Basic residues in the extracellular loops of PagC establish polar contacts with LPS

To gain molecular insights into the role of PagC in OMV production, we performed MD simulations of the protein embedded in a native-like STm OM (inner leaflet: phospholipid, outer leaflet: LPS). The initial structural model of PagC was obtained from the AlphaFold database[31] and then used to initiate three independent MD simulations. During 3 μs of MD simulation, PagC maintains a stable membrane-inserted conformation that optimizes hydrophobic match with the membrane (Fig. 3A) with an average β-barrel transmembrane tilt of 15° (Fig. S5A, B).

The simulations revealed multiple interactions between PagC and LPS. Basic residues extending from the base (K13, R21, K68, K96, R117, and K150) and midsections (K64, K100, and K113) of the extracellular loops form polar contacts (including hydrogen bond and electrostatic interactions) with the lipid A, KDO, inner core and outer core groups of LPS (Fig. S6A) and many of these contacts are long-lived on the time scale of the MD simulation (Fig. S6B). Moreover, acidic residues in the extracellular loops of PagC form metal ion-mediated polar contacts to LPS phosphate groups. Such stable association of the protein with LPS results in the effective formation of a PagC-LPS assembly where on average one protein associates with seven LPS molecules (Fig. S6F). Such interactions are expected to

result in greater order and lower flexibility of both PagC and LPS, as reported previously for Ail[29]. The MD data, therefore, point to a role for outward-facing basic and acidic residues of PagC in stabilizing PagC-LPS complexation, a process that could lead to lateral phase separation of PagC-LPS rafts and membrane remodeling, with ultimate consequences for OMV production.

### The second and third extracellular loops of PagC contain a pH response motif

The experimental data and MD simulations each show that basic residues in the extracellular loops of PagC are important for PagC-mediated OMV formation and for establishing PagC-LPS contacts. However, basic residues alone do not explain why RcK and Ail, which carry similarly placed basic residues (Fig. S4B), do not activate OMV production. Analysis of the three protein sequences revealed a distinguishing feature of PagC: the presence of two amino acid segments with inverse homology and three His residues (H60, H62, H102) in EL2 and EL3. Residues 62–69 in the descending segment of EL2 are homologous to residues 102-95 in the inverse ascending segment of EL3 (Fig. 3B) and their topological arrangement juxtaposes His and Phe sidechains, priming them for complementary ring stacking interactions across the EL2-EL3 interface. Indeed, MD simulations performed for PagC with neutral His sidechains, mimicking neutral pH, show that these complementary segments participate in a series of EL2-EL3 cross contacts (Fig. 3C, D), including H60-H62-H102 and F65-F99 ring stacking. Together these interactions restrain and couple the conformation and dynamics of EL2 and EL3, resulting in similar

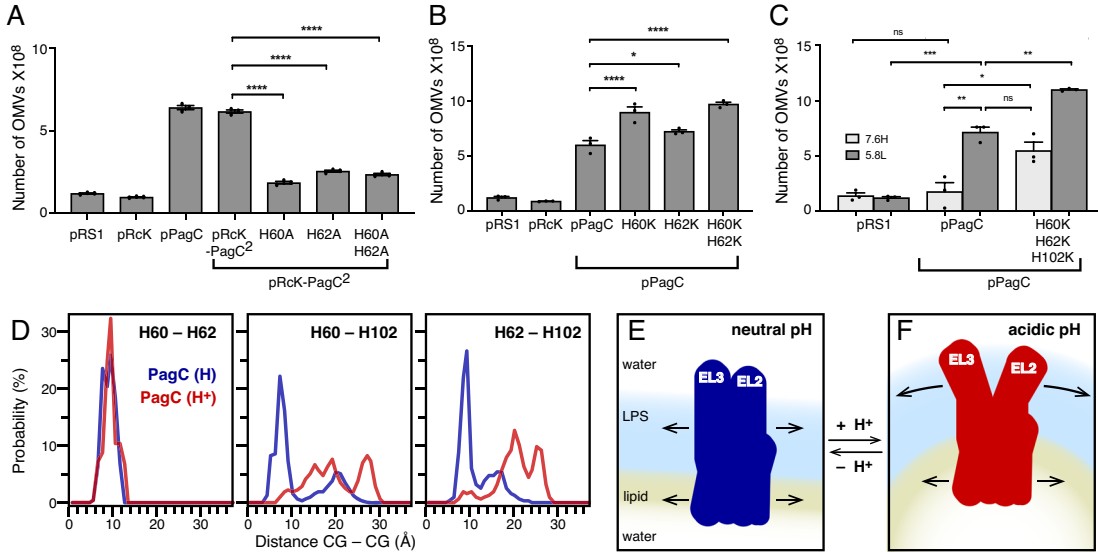

**Fig. 4 | Role of the pH response His in OMV production.** Number of OMVs produced by His mutants in the background of the pRcK-PagC[2] chimera (**A**) or pPagC (**B**, **C**). Bacteria transformed with empty plasmid pRS1 serve as negative control. In (**A**) and (**B**), statistical significance was calculated using a one-way ANOVA multiple-comparison test and set at a $p$ value < 0.05 (*, $p = 0.024$; ****, $p < 0.0001$). In (**C**), statistical significance was calculated using a two-tailed non-parametric unpaired $t$ test and set at a $p$ value < 0.05 (*, $p = 0.030$; **, $p = 0.0012$; ***, $p = 0.0002$; ns, $p > 0.05$). Mean values with SEM are shown ($n = 3$ biological replicates). **D** Probability distribution of His-His distances. Each trace is the average over the last 2 μs of three independent 3 μs MD simulations of PagC with neutral (blue) or protonated (red) His. **E**, **F** Model for PagC-dependent OM vesiculation in response to pH. Source data are provided as a Source Data file.

levels of conformational flexibility, of the two loops (Fig. 3F, Fig. S7A, *blue trace*).

The three His are unique to PagC and particularly interesting because their characteristic pKa (~6) makes them susceptible to protonation and acquisition of a positive charge in the mildly acidic intravacuolar environment encountered by STm upon host cell invasion[18–20]. As acidification is a known activator of PhoPQ and PagC expression[18,19], we examined the OMV-promoting activity of the His residues by mutating H60 and H62 in both the background of the pRcK-PagC[2] chimera and pPagC. Since the MD simulations indicate that H60 and H62 in EL2 each participate in stacking interactions with H102 in EL3, we reasoned that substituting either one would break the His-mediated EL2-EL3 connectivity. Mutation of either His to Ala resulted in substantially lower OMV levels relative to either pPagC or pRck-PagC[2] in 5.8L (pH 5.8) media (Fig. 4A), indicating that H60 and H62 play a critical role in OM vesiculation, and their deactivation reduces OMV production by STm. To mimic acidic conditions where His becomes protonated and positively charged, we mutated His to Lys. In this case, the H60K, H62K, and double H60K, H62K mutations in pPagC-enhanced OMV production relative to the wildtype PagC (Fig. 4B).

To further examine the effect of pH, we analyzed OMV production by a triple His mutant pPagC (H60K, H62K, and H102K) in both 5.8L (pH 5.8) and 7.6H (pH 7.6) media. Mutation of all three His to positively charged Lys resulted in significantly greater OMV production relative to pPagC in both media (Fig. 4C) indicating that the triple His-Lys mutant is constitutively activated for OMV production. Furthermore, the elevated OMV production of the triple His-Lys mutant relative the pPagC in 5.8L may reflect the protonation capacity of His residues in PagC. The pKa values of His in some proteins have been shown to vary between 5.4 and 7.6, thus it is possible that a pH of 5.8 is not sufficiently low to fully protonate all three His residues of PagC and fully replicate the activation observed for triple Lys substitution mutant. In macrophage vacuoles, where pH estimates are in the range of 4.4–5.3[18–20], PagC may be expected to resemble the triple His-Lys mutant more closely. Together, these data support the hypothesis that PagC-mediated vesiculation becomes activated at mildly acidic pH where His becomes protonated.

We explored the molecular basis for this effect by performing MD simulations with His-protonated PagC to mimic the mildly acidic environment. The results of three independent 3 μs MD simulations show that His-protonated PagC remains stably embedded in the OM with a smaller and narrower distribution of transmembrane tilt angle (average 12°) compared to the His-neutral state (Fig. S5A, B). The molecular interaction and conformational flexibility profiles, however, change significantly. His protonation alters the profile of PagC interaction with the OM (Fig. S8) resulting in reduced contacts of the barrel with the lipid and LPS acyl chains, reduced contacts with water, especially for EL2, EL3, and EL4, enhanced contacts of H60 and H62 with the LPS inner core and altered contacts with outer core LPS groups. Moreover, His protonation alters the flexibility profile of EL2 and EL3, leading to both significantly reduced flexibility in the ascending segment of EL2 and greater flexibility in the ascending segment of EL3 (Fig. 3F, S7A, *red trace*). A reduction in dynamics at residue E55 appears to be a common feature of both His-neutral and His-protonated PagC. It is also observed in MD simulations performed in a symmetric phospholipid bilayer without LPS (Fig. S7B), and likely related to long-lived interactions of E55 with the His cluster in both states (Fig. S7C).

Principal component (PC) analysis reveals that the three MD simulations for His-neutral PagC populate distinct regions of the conformational landscape, with only little overlap indicative of limited conformational exchange (Fig. S9A, B, *blue*). Overall, the data indicate that deep low-energy wells, and thus high energy barriers, in the conformational landscape of His-neutral PagC restrain large-scale dynamics and the ability to converge to a single conformation in the timescale of this 3 μs simulation. The situation for His-protonated PagC is distinctly different (Fig. S9A, B, *red*). Here, significant overlap between the three independent MD simulations indicates the protein's capacity to explore the range of its conformational landscape and converge to one overall conformation. These differences are reflected in the conformations of EL2 and EL3. In His-neutral PagC, EL2 and EL3 both adopt more extended conformations that snorkel out of the OM, with similar flexibilities that are mutually restrained by complementation of the inverse-homology pH response motif (Fig. 3C, D;

Fig. S9D) and stabilized by contacts of polar residues with the surrounding LPS. In His-protonated PagC, on the other hand, mutual electrostatic repulsions of the electropositive His cause a large increase in His-His distances across EL2-EL3 (Fig. 4D; S5), net unraveling of EL2-EL3 complementation and decoupling of their motions and conformations (Fig. 3E; Fig. S9E). The charged His also forms greater numbers of long-lived polar contacts with LPS outer core phosphates (Fig. S6C, D) and appears to promote conformations of EL2 and EL3 that are more deeply embedded in the OM. It appears, therefore, that the complementary sequences of EL2 and EL3 can switch conformation and dynamics upon transfer between neutral and acidic environments. Their high sensitivity to small pH alterations is highly suggestive of a pH response motif that enables PagC and STm to adapt to the environmental conditions encountered upon host cell invasion.

## Proposed role of PagC in OMV formation

OMV formation is generally thought to be driven by the insertion of curvature-inducing molecules such as modified LPS into the OM, loosening of OM interactions with the underlying peptidoglycan, and outward turgor pressure resulting from the accumulation of peptidoglycan fragments and misfolded proteins in the periplasmic space[7,9–14]. Our findings suggest a mechanism for control of OM vesiculation mediated by the conformational changes of an outer membrane protein that are stimulated by environmental pH.

Here, we have shown that the PhoPQ-induced OM protein PagC enhances OMV production independently of LPS-modifying chemical activities, such as PagL and PagP. We found that the second and third extracellular loops of PagC are critical for OMV biogenesis and sufficient for conferring OMV production activity in the OMV-inactive homolog RcK. MD simulations unequivocally revealed that basic residues in EL2 and EL3 establish multiple interactions with the surrounding LPS that are expected to stabilize the OM against a variety of stressors as observed for the PagC homolog Ail[29]. Nevertheless, the most striking locus of OM vesiculation activity is situated in a pH-responsive amino acid motif in PagC that harbors three histidine residues. Mutagenesis experiments revealed that replacing these His with Ala suppressed OMV production in STm, while replacing them with positively charged Lys had the opposite effect, generating a constitutively super-activated PagC capable of producing OMV levels greater than wild-type.

These results suggest a model for the way in which PagC senses the pH of its environment and promotes OM vesiculation. At neutral pH (Fig. 4E) His stacking interactions promote aromatic-aromatic and polar contacts that result in dynamically coupled extended conformations of EL2 and EL3 and an overall cylindrical shape of PagC. At acidic pH (Fig. 4F), electropositive His-His repulsions decouple EL2 and EL3, and the protein becomes more deeply embedded in the outer membrane. Acidification, thus may alter the shape and dynamics of PagC from a cylinder at neutral pH, exerting equal lateral pressure on the inner phospholipid and outer LPS leaflets of the OM, to a wedge at acidic pH, exerting more lateral pressure on the OM outer leaflet. While the present MD simulations were not designed to capture membrane curvature and are limited by the lack of an experimental structure of PagC, they indicate that His protonation alters the association of PagC with the inner and outer regions of the OM. Additional long simulations with an extended OM and enhanced sampling will be important for examining the effects of His protonation and His mutations on membrane structure.

His residue protonation plays a central role in pH sensing by proteins in various physiological settings. Viruses that infect via endosomal uptake are known to rely on His-based sensors of local pH to activate protein conformational changes that allow them to fuse with the host membrane and penetrate into the cytosol[32]. His titration in response to local pH can affect the interactions of antimicrobial peptides with membranes[33], and His protonation states are important

for hemoglobin structure and function[34]. In STm, OMV formation appears to be activated by an additive mechanism of pH sensing since PhoPQ is activated by low pH to express the OM protein PagC, which is itself activated by low pH via its His-containing pH response motif to produce more OMVs.

We note that while the His residues provide the pH-responsive switch that appears to activate a cylinder-to-wedge transformation of PagC, their flanking complementary sequences are also likely important. Notably, the native sequence of RcK has Arg residues (R61, R101) at positions analogous to H60 and H102, but it is not active with respect to OMV production. Unlike PagC, however, RcK also contains complementary charge pairs (D87-K122, K89-E121, F42-K71) that may be expected to restrain the loops and constitutively promote the cylindrical protein conformation.

Biophysical studies have shown that the interactions of wedge-shaped proteins with asymmetric membranes can drive membrane curvature and vesicle budding[35,36]. For example, wedge-shaped membrane-spanning helical hairpins and their flanking amphipathic helices can both sense membrane curvature and act as mechanical drivers of asymmetric membrane vesiculation[36]. We propose that similar forces may be active for β-barrel membrane proteins that can alter their shape in the highly asymmetric OM of Gram-negative bacteria. Our work, therefore, provides a perspective on understanding of how specific OM proteins can direct OMV production, a result with potentially important consequences for the development of vaccines and designer OMVs.

## Methods
### Bacterial strains and culture conditions
All bacterial strains including mutants and plasmids used in this study are listed in Table S1. Unless stated otherwise, all reagents used were procured from MilliporeSigma (St. Louis, MO, USA). The STm strains were grown at 37 °C in Lysogeny broth (LB), PhoPQ-inducing pH 5.8, 10 μM $Mg^{2+}$ (5.8L) or non-PhoPQ inducing pH 7.6, 10 mM $Mg^{2+}$ (7.6H) N-minimal media as described previously[25]. To grow STm under PhoPQ-inducing conditions, bacteria were grown in LB and then sequentially transitioned into 7.6H and 5.8L conditions. An STm SL1344 defective in the production of flagellins (*fliC* and *fljB*), designated here as wild-type was used to engineer STm deletion mutants as described previously[25]. The fitness of deletion mutants was confirmed by growth curve analysis and transmission electron microscopy to assay for changes in cell shape/viability. The primers used in this study are listed in Table S1. When appropriate, antibiotics were used at the following concentrations: kanamycin, 25 μg/ml, chloramphenicol, 30 μg/ml, tetracycline, 10 μg/ml and streptomycin, 90 μg/ml.

### PagC chimera and allelic mutant construction
A Δ*pagC*Δ*rck*Δ*ompX*Δ*pgtE* STm quadruple deletion mutant[25] was used to generate PagC-Rck chimeras. Briefly, specific regions of the *pagC* and *rck* genes of STm SL1344 were amplified using primers listed in Table S1. The primers for amplification were designed in such a way that each amplicon carried 13–25 bp upstream and downstream overlaps corresponding to its adjacent chimera region. These amplicons were then fused together to create *pagC-rck* chimeric genes, which were then cloned into the pRS1 plasmid[25] using NEBuilder HiFi DNA assembly master mix. The PagC-H60K + H62K + H102K mutant was created by cloning a PagC-H60K + H62K + H102K gBlock DNA fragment (synthesized by IDT DNA) into pRS1 plasmid. Resulting plasmid DNAs were then transformed into Δ*pagC*Δ*rck*Δ*ompX*Δ*pgtE* mutant and protein expression was induced with anhydrotetracycline hydrochloride (AHT, 0.4 μg/ml) for 2 h at 37 °C. The expression of mutant protein in the bacterial outer membrane was checked by western blotting as described below. The growth of strains harboring PagC-Rck chimeras was very similar to wild-type in LB or N-minimal media.

Substitution of amino acid residues with basic side chains in EL2 and EL3 of PagC was done by site-directed mutagenesis, using the Q5 Site-Directed Mutagenesis Kit (New England Biolabs Inc., Ipswich, MA, USA) according to the manufacturer's protocol. First, the *pagC* gene from *S.* Typhimurium SL1344 genomic DNA was amplified using primers listed in Table S1. Custom mutagenic primers were designed for each mutant, in such a way that the desired nucleotide changes (Arg/Lys/His to Ala) located in the center of the primer with at least 10 complementary nucleotide bases on the 3′ and 5′ end of the primers. Next, an amplicon including 500 bp region upstream and 500 bp downstream region of *pagC* was cloned into pFOK[37] using the NEBuilder HiFi DNA assembly master mix (New England Biolabs Inc., Ipswich, MA, USA). The resulting plasmids were used to replace wild-type *pagC* with *pagC*-containing site specific mutations in the STm genome by allelic exchange methods[37,38]. The sequence of the *pagC* allele mutations were confirmed by DNA sequencing.

## OMV isolation and characterization

OMVs were isolated from broth cultures as described previously[25]. Briefly, bacterial cultures grown in 5.8L or 7.6H N-minimal media, to O.D.$_{600}$ of 0.4–0.6, were centrifuged with a Beckman Avanti J-E centrifuge at $15,000 \times g$ (JLA-16.250 rotor) for 10 min, at 4 °C and the resulting supernatants were filtered through 0.45-μ cellulose membrane filters. Filtered supernatants were then ultracentrifuged at $300,000 \times g$ for 90 min at 4 °C, in a Beckman optima L-90K ultracentrifuge, Type 50.2 Ti rotor. The OMV pellets were resuspended in PBS, filtered through a 0.2-μ filter, and stored at −20 °C until use. In previous studies using this protocol, the purified OMV fractions did not have bacterial contamination or cell lysis byproducts[25]. The concentration and size of purified OMVs (normalized by cfu/ml) were enumerated by nanoparticle tracking analysis (NTA) using ZetaView (Particle Matrix, Meerbusch, Germany) at the Cell Function and Imaging Core, Boston Children's Hospital.

## Lipid A extraction and mass spectrometry analysis

Lipid A extraction and MALDI-TOF MS analysis of the LPS isolated from wild-type or Δ*pagC* bacteria and OMVs was done as described previously[39]. Briefly, wild-type or Δ*pagC* bacteria grown overnight in either LB or N-minimal media (5.8L or 7.6H respectively) were pelleted at 8000 x $g$ for 10 min and a loopful of the bacterial pellet or OMVs (50 μL) was resuspended in 250 μL of 70% isobutyric acid and 150 μL of 1 M ammonium hydroxide. The samples were then heated at 100 °C for 1 h followed by an incubation on ice for 5 min. After centrifugation at $8000 \times g$ for 5 min, the supernatant was collected, diluted with 400 μL endotoxin-free water, flash frozen, and lyophilized overnight. The dried material was then washed once with 1 ml of methanol and reconstituted with 100 μl of a mixture of chloroform:methanol:water in the ratio of 12:6:1 (v/v/v). One microliter of the sample was then spotted on a stainless-steel MALDI plate, allowed to dry in air, and overlaid with 1 μl of 10 mg/mL norharmane matrix (reconstituted in 2:1v/v ratio of chloroform:methanol). Each spot was measured in 300 shot steps for a total of 3000 laser shots using a Bruker Autoflex Speed MALDI-TOF mass spectrometer (Bruker Daltonics, Bremen, Germany) operated in negative-ion reflectron mode. The mass spectra were acquired in the mass range of 1500 to 2400 and calibrated with electrospray ionization (ESI) tuning mix (Agilent Technologies, Santa Clara, CA, USA), as described previously[39]. The mass spectra were processed for smoothing and baseline subtraction using FlexAnalysis, version 3.4 (Bruker Daltonics, Bremen, Germany).

## Gas chromatography-based lipid A analysis

LPS fatty acids from wild-type or Δ*pagC* bacteria and OMVs were converted to fatty acid methyl esters and analyzed by gas chromatography as described previously[40]. Briefly, bacterial cell pellets or 50 μl of OMVs were incubated at 70 °C for 1 h in 500 μl of 90% phenol and

500 μl of water. Samples were then cooled on ice for 5 min and centrifuged at 10,000 $g$ for 10 min. The aqueous layer was then collected and 500 μl of water was added to the lower (organic) layer and incubated again. This process was repeated twice, and all aqueous layers were pooled. Two ml of diethyl ether was added to the harvested aqueous layers, and the mixture was then vortexed and centrifuged at 5000 $g$ for 20 min. The lower (organic) phase was then collected, and 2 ml of ether was added back to the remaining aqueous phase. This process was carried out twice more. The collected organic layer was then frozen and lyophilized overnight. LPS fatty acids were converted to fatty methyl esters, in the presence of 10 μg pentadecanoic acid as an internal standard, with 2 M methanolic HCl (Alltech, Lexington, KY) at 90 °C for 18 h.

## Lipopolysaccharide (LPS) assay

Wild-type or Δ*pagC* bacteria were grown in N-minimal media (5.8 L), to O.D.$_{600}$ = 0.5 and 1.5 ml of the culture was harvested at 10,000 x $g$ for 10 min. The bacterial pellet, resuspended in 250 μl of dissociation buffer (1 M Tris-Cl pH 6.8, 10% glycerol, 2% SDS, and 4% β-mercaptoethanol), was then boiled at 100 °C for 10 min and allowed to cool down at room temperature. To remove any insoluble material, the samples were centrifuged at 12,000 × $g$, for 10 min and the resulting supernatant was then incubated with 30 μg of proteinase K at 55 °C, 2 h. After protein digestion the samples were then separated on 15% SDS-PAGE gels and stained with Pro-Q Emerald 300 Lipopolysaccharide gel stain kit according to the manufacturer's protocol.

## Outer membrane preparations and Western blotting

OM proteins were isolated from 10 ml bacterial culture grown till O.D.$_{600}$ = 0.5 by pelleting the bacteria at 8000 x $g$ for 10 min at 4 °C. The bacterial pellet was then resuspended in 90 μl Lysis Buffer L1 (20% sucrose, 30 mM Tris Cl, pH = 8), 1 mM PMSF and10 μl Lysozyme from a 1 mg/ml stock made in 0.1 M EDTA and incubated on ice for 30 min. Sterile MgCl$_2$ (final concentration 20 mM) was then added to the bacterial suspension to stabilize the spheroplasts and the suspension was then centrifuged at 15000 $g$ for 2 min. Supernatant containing periplasmic proteins was then removed, and the pellet was resuspended in 100 μl Lysis buffer L2 (10 mM Tris-Cl, 100 mM NaCl, 10 mM MgCl$_2$, 1 mM EDTA) containing 1 mM PMSF and 1 μg/ml DNase. The pellet was then sonicated for 10 pulses with 1 min gap at maximum[10] amplitude. 0.5% Sarcosyl was then added to the mixture and incubated at room temperature for 20 min to solubilize the OM. Intact cells or unsolubilized material were removed by pelleting the mixture at 5000 g, for 5 min, and the supernatant was ultracentrifuged in a 50.2 Ti rotor at 50,000 rpm for 90 min to pellet the OM proteins. The isolated OM proteins were separated by SDS-PAGE and electro-transferred to a nitrocellulose membrane for western blotting. The presence of PagC, Rck, PagC-Rck chimeras, PagC-alanine, or lysine substituents in the OM protein fraction was detected by using rabbit anti-PagC antibody at a 1:500 dilution. A goat anti-rabbit HRP conjugated antibody at 1:10,000 dilution was used as the secondary antibody (Jackson Immuno Research, PA, USA) followed by probing the blots with ECL substrate (GE Healthcare Life Sciences, MA, USA). Uncropped blot scans are provided in source data file.

## Bioinformatics

Protein sequences were obtained from the UniProt database (accession numbers: P23988 for PagC; Q04817 for RcK, and A0A5P8YI02 for Ail) and analyzed in Jalview[41]. Initial sequence alignments were generated with ClustalX[42] and then refined by several rounds of manual editing guided by the known eight-stranded β-barrel structure of Ail (PDB: 2N2L) and the structural models predicted from AlphaFold for PagC (model: AF-P23988; accessed: June 01, 2022) and RcK (model: AF-Q04817; accessed: June 01, 2022). Sequence identity, conservation, and consensus were calculated with Jalview.

## Statistics

One-way ANOVA multiple comparisons or two-tailed unpaired *t* test were used for statistical calculations using Prism v. 10 (GraphPad Software, San Diego, CA, USA). Statistical significance was set at $p < 0.05$ (* $p < 0.05$, ** $p < 0.01$, *** $p < 0.001$ and **** $p < 0.0001$).

## MD simulations

All-atom MD simulations were performed using the CHARMM36 force fields for protein, lipids, carbohydrate and LPS[43–46] with the TIP3P water model[47]. Briefly, all systems were prepared using CHARMM-GUI *Membrane Builder*[48–51] and equilibrated with the CHARMM-GUI standard protocol. The temperature was maintained at 310.15K using Langevin dynamics, and pressure was maintained at 1 bar using the semi-isotropic Monte-Carlo barostat method[52,53] with a $5\,ps^{-1}$ of coupling frequency.

A total of eight independent MD simulations was performed, starting from one structural model of PagC, with either neutral or protonated His (Table S2). The starting structure was obtained from the AlphaFold protein structure database (code: AF-P23988-F1[31]) and then embedded in the membrane with different initial values of membrane depth and orientation in the different replicas, so that each MD run represents a distinct independent experiment. Each model was embedded in either an asymmetric STm outer membrane or a symmetric phospholipid lipid bilayer containing DMPC (dimyristoyl-phosphatidylcholine) and DMPG (dimyristoyl-phosphatidylglycerol).

The outer leaflet of the STm OM contained 25 LPS molecules. The structure of STm LPS (Fig. 1A) includes a lipid A moiety with hexa-acylated fatty acid tails[54] The LPS core contained: βD-GlcNAc(1→6)αD-Glc(1→2)αD-Gal(1→3)[αD-Gal(1→6)]αD-Glc(1→3)αLD-Hep(1→3)αLD-Hep(1→5)[αD-Kdo(2→4)]αD-Kdo(2→, where D-GlcNAc is D-glucosamine, D-Glc is D-glucose, D-Gal is D-galactose, LD-Hep is L-glycero-D-manno-Heptose, and D-Kdo is D-3-deoxy-D-manno-oct-2-ulosonic acid. The inner leaflet of the STm OM contained a 3:1 molar ratio of DMPC to DMPG. $Ca^{2+}$ counterions were added to neutralize the LPS charges, and bulk 150 mM KCl was added to mimic the biological ionic strength.

MD production simulations were conducted with OpenMM[55] with a time step interval of 2 fs. The six simulations of PagC in STm OM were conducted for $3\,\mu s$ and the last $2\,\mu s$ of trajectories were used for analysis, while the two simulations of PagC in symmetric lipid bilayer were conducted for $1.5\,\mu s$ and the last $1\,\mu s$ used for analysis. For each simulation, poses were saved every 0.5 ns of the trajectory duration, and trajectories from all simulations were combined for analysis in each of the three systems. Statistical analyses were performed with home-made Python scripts in MDAnalysis[56], PyLipid[57], and JupyterNotebook.

## Reporting summary

Further information on research design is available in the Nature Portfolio Reporting Summary linked to this article.

## Data availability

Source data for all figures presented in the paper and Supplementary Information are available. AlphaFold structural models: see Supplementary Data. MD data: see Supplementary Data 1. 2N2L: https://www.rcsb.org/structure/2N2L Mass Spectrometry data: https://doi.org/10.5281/zenodo.12734941 Source data are provided with this paper.

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

## Acknowledgements

This work was supported by grants AI139982 (D.M.S); AI042347 and HHMI (M.K.W); GM118186 (F.M.M.) and AI147314 (R.K.E). The funders had no role in the design of the study; in the collection, analyses, or interpretation of data; in the writing of the manuscript, or in the decision to publish the results. We thank Dirk Bumann for plasmids, Tomoko Yamamoto for the PagC antisera, Ye Tian, Kyungsoo Shin, and Wonpil Im for assistance with the MD simulations, and Waldor lab members for comments on manuscript. This article is subject to HHMI's Open Access to Publications policy. HHMI lab heads have previously granted a non-exclusive CC BY 4.0 license to the public and a sublicensable license to HHMI in their research articles. Pursuant to those licenses, the author-accepted manuscript of this article can be made freely available under a CC BY 4.0 license immediately upon publication.

## Author contributions

R.D., D.M.S., R.K.E., F.M.M., and M.K.W. were involved in research design and discussion of results; R.D., T.G., and R.D.S. performed experiments;

R.D., T.G., and F.M.M. analyzed data; R.D., F.M.M., and M.K.W. wrote the manuscript with input and approval from all authors.

## Competing interests

The authors declare no competing interests.
