## [Peer Review File · Nature Communications]

A pH-sensitive motif in an outer membrane protein activates bacterial membrane vesicle productionReviewer #1 (Remarks to the Author):

Dehinwal et al. report a novel mechanism for outer membrane vesicle (OMV) production in *Salmonella Typhimurium* (STm). Through a combination of mutagenesis, lipidomics, and cell-level assays combined with structural insights from molecular dynamics (MD) simulations, the authors propose that pH-dependent conformational changes in the extracellular loops (EL) of the outer membrane protein (OMP) PagC associated with the protonation state of three His residues. Upon His protonation, PagC would assume an overall wedge-like shape resulting in a pressure differential between lipids that would eventually lead to vesiculation. The authors have previously shown that PagC is key to OMV production in STm and the present contribution offers compelling structural insights that were previously missing. Direct structural evidence comes entirely from simulations, but the role of the His residues (and thus the pH effect) are a prediction from the MD simulations validated by further experiments, which strengthens the authors' hypothesis. At the same time, the lack of direct experimental structural evidence mandates a close scrutiny of the MD results. Furthermore, the authors have chosen to include some "mechanistic cartoons" (cf. Fig 4F-G) that are, in my opinion, not entirely justified by the MD simulation data as presented. From the presentation, it is also not entirely clear how these conformational changes relate to membrane budding and vesiculation. I believe a revised version of this contribution will be of interest to the broad membrane/cell biophysics community and could have significant implications several areas of microbiology. It should be considered for publication provided the authors are able to bridge the gap between their actual simulation data and their very appealing conclusions. The specific points requiring attention are indicated below:

1. According to Fig. 4 and accompanying text, upon His protonation (acidic conditions) EL2 and EL3 change conformation from fully extended in the His-neutral case to a "collapsed" (disordered?) conformation. The authors' claim is that through these conformational changes the protein assumes a wedge-like shape with stronger interactions with the outer leaflet of the OM. However the associated data presented from the MD simulations is not very meaningful or easy to read.

1a) Protein conformation: The separation between His residues is clear, but the only actual structural data offered for the entire protein or EL regions are barrel tilt evolution and distributions (fig S4), radius of gyration distributions (fig 4), and mean-squared fluctuations (fig 3) all of which suggest only modest changes in conformational dynamics, secondary structure, or tertiary structure. If the ELs are changing secondary structure as dramatically as illustrated by the Fig 4 cartoons, then a per-residue secondary structure assignment per residue as a function of time (e.g. using DSSP or STRIDE) should show these changes very clearly. Regarding the difference between a "cylindrical shape" and a "wedge shape", I'd suggest looking at the evolution and distribution of principal moments. Another issue with the conformational dynamics data as presented is that, for the most part, the authors aggregate the sampling from three independent simulations without showing that these are indeed sampling from the same distributions. It is critical the authors show that either their three independent trajectories sample for the same distribution or that they sample from separate but relevant portions of the EL conformational landscape. This could be achieved by monitoring key backbone dihedrals throughout the sequence. On a final note, it is unclear what the purpose of Fig S4A is, the mean-squared deviations from the initial configuration are likely dominated by the TM barrel, whether or not these data is relevant to the ELs is unclear and perhaps misleading since the reference (AlphaFold prediction) is likely not accurate in the EL region.

1b) protein-membrane interactions: there are two key points associated to the authors proposed mechanism that require a detailed presentation of the interaction between the protein and its environment: First, if the His are to change protonation state they should be accessible to solvent in both neutral and acidic conditions. This should be shown explicitly. The current presentation only emphasizes inter-residue interactions. Second, if we are to be persuaded that the interactions between the ELs and the membrane found in the present simulation could potentially lead to membrane curvature, we need a clearer account than Fig S5. I'd suggest accompanying these figures with a scheme of the OM illustrating the different regions plus details of the interactions that are specific to either protonation state including their time evolution. If this is the purpose of Fig. S6, then it is unclear why isn't there a contrast between His-neutral and His-protonated. Membrane thickness maps around the OMP as those reported in ref. 30 (Singh et al) may be useful

here as well. Here again, the author should be careful to show that their treatment of the three independent simulations (as sampling from the same distribution) is justified.

2. The authors indicate that the same MD simulation protocols as those in ref. 30 (Singh et al) were followed. We can take the results in ref. 30, where the simulations are contrasted to structural experimental data as unambiguous validation of the MD protocol (e.g. force field accuracy). At the same time those simulations were run for 1.5 microseconds. Thus, the question arises of whether or not three independent 500-ns simulations are adequate to sample the relevant dynamics of the ELs. One way to address this matter is performing principal component analysis of the protein backbone covariance matrix and verify that together the His-protonated and His-neutral trajectories sample the relevant dynamics adequately (cf. Hess Physical Review E, 65, 031910).

3. The Methods section and Table S2 refer to His-neutral simulations performed in a symmetric phospholipid bilayer. As far as I can tell the results from these simulations are not reported anywhere. If these simulations do exist, I believe their results would strengthen the authors' case as meaningful negative control. Of course, it would be even better if similar simulations could be performed for His-protonated.

Minor: "mean-squared displacement" is referred to "mean-squared fluctuation" in the caption of Fig. S4. Also, I'd suggest these results, as well as those in Fig. 3, be expressed as "root mean-squared" as they'd be easier to read.

Reviewer #2 (Remarks to the Author):

In this manuscript, Dehinwal and colleagues investigated the molecular mechanism of increased OMV production governed by the PhoP-activated gene PagC in *Salmonella Typhimurium*. By structural analysis, molecular dynamics simulations and vesicle quantification performed in mutants under different pH conditions, the authors showed that the outer membrane protein PagC changes its protonation state at the level of the extracellular loops EL2 and EL3. Changes in protonation at defined His residues modify the conformational state of EL2 and EL3, impacting on the protein conformational state (cylindrical vs wedged) and its interaction with LPS, ultimately affecting the membrane curvature and, subsequently, vesicle production. For this reviewer, the manuscript is properly conceived and elegantly executed. However, major concerns must be addressed:

1- The authors must include lysis controls to ensure that the purified OMV fractions do not contain lysis byproducts. For example: Western blot for the purified OMV fractions employing antibodies against abundant cytoplasmic proteins, such as: ribosomal proteins (EF-TU), chaperones (GroEL, GroES, DnaK), RpoB, metabolism-related proteins, etc.

2- In line with the previous comment: the authors are employing a quadruple mutant in a flagellin deficient background strain. All these mutations delete OM proteins, which might have an impact on membrane stability. Growth curves and TEM images must be included for the employed strain as well as Stm WT (for comparative purposes) to analyze general cell shape and viability and rule out the possibility of cell lysis, which is essential in the context of analyzing OMV production.

Comments:

- Lane 49: "subsp". and "serovar Typhimurium" should not be italicized. Please modify.
- Lane 82 (related to figure 1D): The authors should also show the sugar profiling for OMVs fractions.
- Lane 114: Authors are employing a quadruple mutant in a flagellin mutant background. Growth curves and TEM images should be included for this strain as well as the WT to analyze general cell viability and rule out the possibility of cell lysis, essential in the context of analyzing bonafide OMVs.
- Lane 115: Which relevant information is providing the WB in figure 2A for the chimeric proteins?

- Lane 132: Please indicate how the representation in figure 2C was made.
- Lane 136: As indicated in the figure 2C, R46 and K118 are in-facing residues. Please modify accordingly.
- Lane 147 (related to figure S3A): Are those different PagC mutants equally packed into OMVs? Just curious...
- Lane 154 (related to figure 3A): For this reviewer, the figure needs more detailed explanation about the meaningful interactions to make a better understanding. What is figure 3A trying to highlight? Please indicate what the rainbow color transition means in the beta barrel structure. Please also indicate the relevant interactions and why the 342 ns snapshot was selected. Some of these inquiries are explained in the figure legend after C-D, but in my opinion they should appear before.
- Lane 318: "fused" or "recombined" could be used instead of "stitched".
- Lane 338: For how long were the cultures grown? In which growth phase/OD were the cultures harvested for OMV isolation?
- Lane 422: Why do the authors choose 25 LPS molecules?

Reviewer comments are listed (*in italics*) and addressed individually below (in blue).

Reviewer #1:

1. According to Fig. 4 and accompanying text, upon His protonation (acidic conditions) EL2 and EL3 change conformation from fully extended in the His-neutral case to a “collapsed” (disordered?) conformation. The authors’ claim is that through these conformational changes the protein assumes a wedge-like shape with stronger interactions with the outer leaflet of the OM. However the associated data presented from the MD simulations is not very meaningful or easy to read.

Response: Thank you for the thoughtful comments. As outlined below, we have extended the MD simulations from 500 ns to 3,000 ns and carried out extensive additional analyses that support our model. The text and figures related to the MD section of the manuscript have been revised to reflect the new analyses.

1a) Protein conformation: The separation between His residues is clear, but the only actual structural data offered for the entire protein or EL regions are barrel tilt evolution and distributions (fig S4), radius of gyration distributions (fig 4), and mean-squared fluctuations (fig 3) all of which suggest only modest changes in conformational dynamics, secondary structure, or tertiary structure. If the ELs are changing secondary structure as dramatically as illustrated by the Fig 4 cartoons, then a per-residue secondary structure assignment per residue as a function of time (e.g. using DSSP or STRIDE) should show these changes very clearly. Regarding the difference between a “cylindrical shape” and a “wedge shape”, I’d suggest looking at the evolution and distribution of principal moments. Another issue with the conformational dynamics data as presented is that, for the most part, the authors aggregate the sampling from three independent simulations without showing that these are indeed sampling from the same distributions. It is critical the authors show that either their three independent trajectories sample for the same distribution or that they sample from separate but relevant portions of the EL conformational landscape. This could be achieved by monitoring key backbone dihedrals throughout the sequence. On a final note, it is unclear what the purpose of Fig S4A is, the mean-squared deviations from the initial configuration are likely dominated by the TM barrel, whether or not these data is relevant to the ELs is unclear and perhaps misleading since the reference (AlphaFold prediction) is likely not accurate in the EL region.

Response: We have extended the MD simulations from 500 ns to 3,000 ns and have performed principal component (PC) analysis to examine the structural and dynamic properties of PagC. The main manuscript and supporting information have been revised to present the new data and analyses.

Regarding secondary structure, the data point to a conformational change where the two major loops EL2 and EL3 become decoupled and splay apart in the His-protonated state of PagC, although they do not significantly alter their secondary structures reflected in the backbone dihedral angles.

Regarding sampling, the extended data and PC analyses show that the three independent trajectories of His-neutral PagC sample from separate but relevant portions of the EL2 and EL3 conformational landscape, while the three independent trajectories of His-protonated PagC sample a common conformational region. The data reflect a net unraveling of EL2-EL3 complementation and decoupling of their motions and conformations. This is discussed in the revised manuscript text related to **Fig. S8**.

We have revised this figure (**Fig. S4**) and no longer show the time evolution of RMSD from the initial model.

1b) protein-membrane interactions: there are two key points associated to the authors proposed mechanism that require a detailed presentation of the interaction between the protein and its environment: First, if the His are to change protonation state they should be accessible to solvent in both neutral and acidic conditions. This should be shown explicitly. The current presentation only emphasizes inter-residue interactions. Second, if we are to be persuaded that the interactions between the ELs and the membrane found in the present simulation could potentially lead to membrane curvature, we need a clearer account than Fig S5. I’d suggest accompanying this figure with a scheme of the OM illustrating the different regions plus details of the interactions that are specific to either protonation state including their time evolution. If this is the purpose of Fig. S6, then it is unclear why isn’t there a contrast between His-neutral and His-protonated. Membrane thickness maps around the OMP as those reported in ref. 30 (Singh et al) may be useful here as well. Here

again, the author should be careful to show that their treatment of the three independent simulations (as sampling from the same distribution) is justified.

Response: The map of atomic contacts has been updated and now reflects time-averages over the 3 μ s simulations (Fig. S7). The map shows that the His residues are water-accessible. The revised manuscript also provides a detailed analysis of all polar contacts between PagC Arg, Lys, and His sidechains to LPS for both His-neutral and His-protonated states (Fig. S5).

2. The authors indicate that the same MD simulation protocols as those in ref. 30 (Singh et al) were followed. We can take the results in ref. 30, where the simulations are contrasted to structural experimental data as unambiguous validation of the MD protocol (e.g. force field accuracy). At the same time those simulations were run for 1.5 microseconds. Thus, the question arises of whether or not three independent 500-ns simulations are adequate to sample the relevant dynamics of the ELs. One way to address this matter is performing principal component analysis of the protein backbone covariance matrix and verify that together the His-protonated and His-neutral trajectories sample the relevant dynamics adequately (cf. Hess Physical Review E, 65, 031910).

Response: Done. We have extended the MD simulations from 500 ns to 3,000 ns and have performed principal component (PC) analysis to examine the structural and dynamic properties of PagC. Please see our response to comment (1a). While it is impossible to conclude that the MD simulations have reached thermodynamic equilibrium, the data indicate that key conformational and dynamics properties converge (His-His distances, LPS contacts, atomic contacts, loop dynamics and conformational exchange) to shed light on differences between His-neutral and His-protonated states of PagC.

3. The Methods section and Table S2 refer to His-neutral simulations performed in a symmetric phospholipid bilayer. As far as I can tell the results from these simulations are not reported anywhere. If these simulations do exist, I believe their results would strengthen the authors' case as meaningful negative control. Of course, it would be even better if similar simulations could be performed for His-protonated.

Response: The revised manuscript (Fig. S4, S6, and related text) presents the data obtained for two independent 1.5 μ s simulations of His-neutral PagC in a symmetric lipid bilayer without LPS.

Minor: "mean-squared displacement" is referred to "mean-squared fluctuation" in the caption of Fig. S4. Also, I'd suggest these results, as well as those in Fig. 3, be expressed as "root mean-squared" as they'd be easier to read.

Response: Done.

Reviewer #2:

In this manuscript, Dehinwal and colleagues investigated the molecular mechanism of increased OMV production governed by the PhoP-activated gene PagC in Salmonella Typhimurium. By structural analysis, molecular dynamics simulations and vesicle quantification performed in mutants under different pH conditions, the authors showed that the outer membrane protein PagC changes its protonation state at the level of the extracellular loops EL2 and EL3. Changes in protonation at defined His residues modify the conformational state of EL2 and EL3, impacting on the protein conformational state (cylindrical vs wedged) and its interaction with LPS, ultimately affecting the membrane curvature and, subsequently, vesicle production. For this reviewer, the manuscript is properly conceived and elegantly executed. However, major concerns must be addressed:

Response: Thank you for the thoughtful read and positive comments.

1. The authors must include lysis controls to ensure that the purified OMV fractions do not contain lysis byproducts. For example: Western blot for the purified OMV fractions employing antibodies against abundant cytoplasmic proteins, such as: ribosomal proteins (EF-TU), chaperones (GroEL, GroES, DnaK), RpoB, metabolism-related proteins, etc.

Response: We purified OMVs using an established protocol that has been shown to yield OMV fractions free of lysis byproducts. In our previous manuscript (mBio 12:10.1128/mbio.00869-21), where we used the same protocol as the current paper, we analyzed the OMV fraction by TEM and did not observe any cell lysis/debris in the purified OMV fractions analyzed. Additionally, we analyzed our purified OMV fractions by mass spectrometry to profile the protein content (Table S2, mBio 12:10.1128/mbio.00869-21) and did not observe any cytoplasmic proteins contamination. Furthermore, papers that used the same protocol we used (including <https://doi.org/10.3389/fmicb.2017.00134> and <https://doi.org/10.1016/j.micres.2015.06.012>;) demonstrated that there is no detectable GroEL in the purified OMV fraction by western blotting.

2. In line with the previous comment: the authors are employing a quadruple mutant in a flagellin deficient background strain. All these mutations delete OM proteins, which might have an impact on membrane stability. Growth curves and TEM images must be included for the employed strain as well as Stm WT (for comparative purposes) to analyze general cell shape and viability and rule out the possibility of cell lysis, which is essential in the context of analyzing OMV production.

Response: Thanks for the suggestion. As shown below (Fig. R1), the growth curve and shape (by TEM) of the quadruple mutant and WT are indistinguishable.

Fig R1: (A) Growth curve of Wildtype (WT), $\Delta pagC$ and $\Delta pagC\Delta rck\Delta ompX\Delta pgtE$ STm grown in LB media. Optical density at 600 nm (OD₆₀₀) was measured every 10 mins for 24 hours at 37°C, 180 rpm. Results represent three biological replicates. (B) Four representative images of WT and $\Delta pagC\Delta rck\Delta ompX\Delta pgtE$ STm captured by Transmission electron microscopy (TEM). The bacteria were grown in LB media or PhoPQ inducing 5.8L media, stained with 1% aqueous uranyl acetate and viewed under Tecnai 12 electron microscope.

Comments:

- Lane 49: “*subsp.*” and “*serovar Typhimurium*” should not be italicized. Please modify.

Response: Done

- Lane 82 (related to figure 1D): The authors should also show the sugar profiling for OMVs fractions.

Response: We suppose that by sugar profiling the reviewer is asking about LPS-carbohydrates. We followed a standard protocol to isolate LPS from WT and PagC mutant (doi.org/10.1073/pnas.120290810). LPS staining with Pro-Q Emerald 300 Lipopolysaccharide gel stain is a highly sensitive method for staining LPS in gels, where the dye binds to periodate-oxidized carbohydrates of LPS and can stain as little as 200 pg of LPS (DOI: [10.1002/1615-9861\(200107\)1:7<841::AID-PROT841>3.0.CO;2-E](https://doi.org/10.1002/1615-9861(200107)1:7<841::AID-PROT841>3.0.CO;2-E)). The appearance of LPS profiles between WT and PagC mutant in Fig 1D did not show any significant difference between the two samples; we think a detailed profiling of the sugar composition of the LPS is beyond the scope of the current study.

- Lane 114: Authors are employing a quadruple mutant in a flagellin mutant background. Growth curves and TEM images should be included for this strain as well as the WT to analyze general cell viability and rule out the possibility of cell lysis, essential in the context of analyzing bonafide OMVs.

Response: See response to point #2 and Figure R1 above.

- Lane 115: Which relevant information is providing the WB in figure 2A for the chimeric proteins?

Response: The Fig 2A WB shows that the chimeric proteins localize to outer membrane thus demonstrating that despite the modifications the chimeric proteins are properly targeted.

- Lane 132: Please indicate how the representation in figure 2C was made.

Response: Figure 2C was made by threading the sequence of PagC onto the predicted eight-stranded β -barrel topology, as defined in the excellent review by Schulz (*Biochim Biophys Acta* 2002, 1565: 308). The Figure was generated with Adobe Illustrator.

- Lane 136: As indicated in the figure 2C, R46 and K118 are in-facing residues. Please modify accordingly.

Response: Done. The text has been corrected.

- Lane 147 (related to figure S3A): Are those different PagC mutants equally packed into OMVs? Just curious.

Response: This is an interesting question that we plan to address in future. We do not know if the mutants are equally packed into OMVs but think that a single residue mutation should not affect the protein expression/packaging considerably.

- Lane 154 (related to figure 3A): For this reviewer, the figure needs more detailed explanation about the meaningful interactions to make a better understanding. What is figure 3A trying to highlight? Please indicate what the rainbow color transition means in the beta barrel structure. Please also indicate the relevant interactions and why the 342 ns snapshot was selected. Some of these inquiries are explained in the figure legend after C-D, but in my opinion they should appear before.

Response: We have removed the rainbow color and revised the Figure and legend. The structure of PagC is now represented in blue for neutral His, or red for protonated His. Fig. 3A is a representative structure taken from the MD simulation trajectory at the end of the simulation (3 μ s) that shows how PagC embeds in the outer membrane.

- Lane 318: “fused” or “recombined” could be used instead of “stitched”.

Response: Done

- Lane 338: For how long were the cultures grown? In which growth phase/OD were the cultures harvested for OMV isolation?

Response: The bacteria were grown for 4 hours, to log phase O.D.₆₀₀ of 0.4-0.6. This has been added to the methods section.

- Lane 422: Why do the authors choose 25 LPS molecules?

Response: The number of lipids and LPS molecules is defined by the size of the volume of the MD simulation setup, to satisfy physical parameters of the simulation.

Reviewer #1 (Remarks to the Author):

In the current version of the manuscript, the authors report extended molecular dynamics simulation trajectories and detailed analysis both of which are sufficient to support their mechanistic model for PagC-mediated OM vesiculation upon acidification. All my comments have been adequately addressed. I believe this revised version of the manuscript will be of interest to the broad membrane/cell biophysics community and should be considered for publication.

Just a minor comment. The following phrase in Methods (747-748) requires clarification: "For each simulation, trajectories were generated every 0.5 ns" perhaps the authors meant to indicate that configuration were saved every 0.5 ns?

Reviewer #2 (Remarks to the Author):

Comments:

1. Please include in the manuscript a phrase indicating that a previous analysis of OMV fractions by mass spectrometry revealed the absence of cell lysis and cite the published paper where it is shown this.
2. Please include data referred as "Fig. R1" as supplementary information.

- Lane 82 (related to figure 1D): I do not agree with the authors when they state that the OMV LPS sugar composition is beyond the scope of the current manuscript. The authors are proposing a mechanism for the increased vesicle production phenotype observed in a mutant and analyzing if there exist modifications in the LPS. As specified in lane 79 of the manuscript, the authors want to examine whether PagC chemically modifies Stm LPS. In first place, the authors analyze the existence of lipid A modifications in the cell (figure 1B), and in vesicles (figure 1C). Subsequently, the authors evaluate the cell LPS profile (figure 1D). Even though the appearance of LPS profiles between WT and PagC mutant did not show any significant difference between the two samples, it seems that the LPS quantity is lower in the mutant. Based on all the previously exposed, it is relevant to analyze whether the LPS profiling is the same in OMVs for both strains to conclude that PagC does not chemically modify LPS.

- Lane 115 (western blot in figure 2A): I believe the western blot is disorienting because it selectively recognizes the portion of the protein the authors are swapping. It should be removed or moved to supplementary information.

Reviewer #1 (Remarks to the Author):

In the current version of the manuscript, the authors report extended molecular dynamics simulation trajectories and detailed analysis both of which are sufficient to support their mechanistic model for PagC-mediated OM vesiculation upon acidification. All my comments have been adequately addressed. I believe this revised version of the manuscript will be of interest to the broad membrane/cell biophysics community and should be considered for publication.

Just a minor comment. The following phrase in Methods (747-748) requires clarification:
"For each simulation, trajectories were generated every 0.5 ns" perhaps the authors meant to indicate that configuration were saved every 0.5 ns?

We have revised the Methods text in question to read "For each simulation, poses were saved every 0.5 ns of the trajectory duration". (line 447)

Reviewer #2 (Remarks to the Author):

Comments:

1. Please include in the manuscript a phrase indicating that a previous analysis of OMV fractions by mass spectrometry revealed the absence of cell lysis and cite the published paper where it is shown this.

We now include a line in methods stating that OMV fraction analysis by mass spectrometry revealed the absence of cell lysis along with the citation. (line 357-358)

2. Please include data referred as "Fig. R1" as supplementary information.

We now include Fig. R1 as supplementary Fig. S3.

- Lane 82 (related to figure 1D): I do not agree with the authors when they state that the OMV LPS sugar composition is beyond the scope of the current manuscript. The authors are proposing a mechanism for the increased vesicle production phenotype observed in a mutant and analyzing if there exist modifications in the LPS. As specified in lane 79 of the manuscript, the authors want to examine whether PagC chemically modifies Stm LPS. In first place, the authors analyze the existence of lipid A modifications in the cell (figure 1B), and in vesicles (figure 1C). Subsequently, the authors evaluate the cell LPS profile (figure 1D). Even though the appearance of LPS profiles between WT and PagC mutant did not show any significant difference between the two samples, it seems that the LPS quantity is lower in the mutant. Based on all the previously exposed, it is relevant to analyze whether the LPS profiling is the same in OMVs for both strains to conclude that PagC does not chemically modify LPS.

We agree that analyses of the sugar content of LPS in OMVs would allow us to draw a definitive conclusion about PagC's potential involvement in LPS modifications. However, this experiment is presently not technically feasible. The high quantity of OMVs required for sugar profiling analyses poses a significant technical challenge because the *pagC* mutant makes only a very small number of OMVs under the requisite *phoPQ*-inducing growth conditions, thus, greatly limiting the feasibility of this experiment.

- Lane 115 (western blot in figure 2A): I believe the western blot is disorienting because it selectively recognizes the portion of the protein the authors are swapping. It should be removed or moved to supplementary information.

The western blot from Fig. 2A has been removed.

We thank the reviewers for their invaluable contributions to our work and hope our current revisions render our manuscript acceptable for publication in *Nature Communications*.